# Saliva molecular testing bypassing RNA extraction is suitable for monitoring and diagnosing SARS-CoV-2 infection in children

**Marta Alenquer**[1], **Tiago Milheiro Silva**[2°], **Onome Akpogheneta**[3°], **Filipe Ferreira**[1], **Sílvia Vale-Costa**[1], **Mónica Medina-Lopes**[1], **Frederico Batista**[4], **Ana Margarida Garcia**[2], **Vasco M. Barreto**[5], **Cathy Paulino**[6], **João Costa**[6], **João Sobral**[6], **Maria Diniz-da-Costa**[6], **Susana Ladeiro**[6], **Rita Corte-Real**[7], **José Delgado Alves**[4,5], **Ricardo B. Leite**[6], **Jocelyne Demengeot**[3], **Maria João Rocha Brito**[2], **Maria João Amorim**[1]*

1 Cell Biology of Viral Infection Lab, Instituto Gulbenkian de Ciência, Oeiras, Portugal, 2 Pediatric Infectious Disease Unit, Hospital Dona Estefânia, Centro Hospitalar Universitário Lisboa Central, Lisboa, Portugal, 3 Lymphocyte Physiology Lab, Instituto Gulbenkian de Ciência, Oeiras, Portugal, 4 Department of Medicine 4, Hospital Prof Doutor Fernando Fonseca, Amadora, Portugal, 5 CEDOC NOVA, Centro de Estudos de Doenças Crónicas, Nova Medical School, Universidade Nova de Lisboa, Lisboa, Portugal, 6 Genomics Unit, Instituto Gulbenkian de Ciência, Oeiras, Portugal, 7 Molecular Biology Laboratory, Department of Clinical Pathology, Centro Hospitalar Universitário Lisboa Central, Lisboa, Portugal

° These authors contributed equally to this work.
* mjamorim@igc.gulbenkian.pt

**Data Availability Statement:** All relevant data are within the paper and its Supporting Information files.

## Abstract

### Background

Adults are being vaccinated against SARS-CoV-2 worldwide, but the longitudinal protection of these vaccines is uncertain, given the ongoing appearance of SARS-CoV-2 variants. Children remain largely unvaccinated and are susceptible to infection, with studies reporting that they actively transmit the virus even when asymptomatic, thus affecting the community.

### Methods

We investigated if saliva is an effective sample for detecting SARS-CoV-2 RNA and antibodies in children, and associated viral RNA levels to infectivity. For that, we used a saliva-based SARS-CoV-2 RT-qPCR test, preceded or not by RNA extraction, in 85 children aged 10 years and under, admitted to the hospital regardless of COVID-19 symptomatology. Amongst these, 29 (63.0%) presented at least one COVID-19 symptom, 46 (54.1%) were positive for SARS-CoV-2 infection, 28 (32.9%) were under the age of 1, and the mean (SD) age was 3.8 (3.4) years. Saliva samples were collected up to 48 h after a nasopharyngeal swab-RT-qPCR test.

### Results

In children aged 10 years and under, the sensitivity, specificity, and accuracy of saliva-RT-qPCR tests compared to NP swab-RT-qPCR were, respectively, 84.8% (71.8%–92.4%), 100% (91.0%–100%), and 91.8% (84.0%–96.6%) with RNA extraction, and 81.8% (68.0%–90.5%), 100% (91.0%–100%), and 90.4% (82.1%–95.0%) without RNA extraction. Rescue

**Funding:** This work was funded by the Fundação para a Ciência and Tecnologia (FCT, Portugal) under RESEARCH4COVID 19 call with reference 283_596885654 and co-funded by ANI under INOV4COVID (Funding to V.M.B.). M.J.A. is funded by the FCT (CEECIND/02373/2020). M.A. is funded by a Junior Researcher working contract from FCT and Instituto Gulbenkian de Ciência (IGC, Portugal). This work benefited from COVID19 emergency funds 2020 from Calouste Gulbenkian Foundation and from Oeiras city council. The funders had no role in study design, data collection and analysis, decision to publish, or preparation of the manuscript.

**Competing interests:** The authors have declared that no competing interests exist.

of infectious particles from saliva was limited to CT values below 26. In addition, we found significant IgM positive responses to SARS-CoV-2 in children positive for SARS-CoV-2 by NP swab and negative by saliva compared to other groups, indicating late infection onset (>7–10 days).

## Conclusions

Saliva is a suitable sample type for diagnosing children aged 10 years and under, including infants aged <1 year, even bypassing RNA extraction methods. Importantly, the detected viral RNA levels were significantly above the infectivity threshold in several samples. Further investigation is required to correlate SARS-CoV-2 RNA levels to viral transmission.

## Introduction

The COVID-19 pandemic caused by SARS-CoV-2 has resulted in excess morbidity and mortality worldwide, especially in elderly populations and people with associated specific comorbidities [1–3]. A great fraction of adults and children above 12 years of age are already fully vaccinated worldwide [4], but younger children remain largely unvaccinated and, therefore, more susceptible to infection. How new variants will be transmitted amongst children and affect the community remains unclear. In fact, thus far, the role of children in SARS-CoV-2 transmission remains poorly understood, mostly because the majority of children with SARS-CoV-2 display mild to no symptoms [5]; it has been estimated that 93% of infected children are not identified by symptom screening [6]. However, it is now well established that children are susceptible to infection [7–9], and a small percentage may develop serious complications [10], including pneumonia, myocarditis, central nervous system disorders, and multisystem inflammatory syndrome [11, 12]. It is critical in this new phase of the pandemic to have readily available strategies for minimally-invasive approaches to monitor school settings; these strategies could help establish whether children are prone to evolve new variants and how variants impact viral transmission by children to the community, by combining diagnosis with genotyping and epidemiological analyses. Saliva molecular testing has emerged as a suitable alternative to nasopharyngeal (NP) swabs for identification of SARS-CoV-2 RNA in children and adults, and for genotyping SARS-CoV-2 variants [13–15]. Compared to NP swabs, saliva testing is less invasive and may be implemented in self-collected or parent-assisted contexts more easily, including the sampling of children up to 1-year-old, by low-pressure aspiration. Saliva RT-qPCR testing is also more sensitive than antigen testing [16]. However, there is still some resistance to using saliva coupled to molecular testing partly because it is not a fast test, and it is unclear if the detected viral loads are substantially lower than those detected in antigen tests using NP swab samples. Importantly, recent studies have demonstrated the potential for equal or higher viral loads in saliva as compared to NP swabs [17]. Furthermore, faster extraction-free saliva tests have shown promising results in adult individuals, but these protocols have not been validated in children [18, 19]. In this cross-sectional study, we focused on children aged 10 years and under, admitted to hospital with COVID-19 related symptoms or with unrelated medical pathologies or surgeries, and investigated the potential of saliva for being coupled to RT-PCR testing, with saliva collected up to 48 h from a positive (or control) NP swab. Interestingly, we found non-significant differences between methods using and bypassing RNA extraction. In addition, we associated RNA levels detected in saliva with infectivity and quantified specific SARS-CoV-2 spike and receptor-binding-domain (RBD) antibodies in this type of

sample. Interestingly, we only found statistically significant differences in IgM levels in samples positive in swab-molecular testing that were negative by saliva-molecular testing, suggesting that discrepancies between saliva and NP molecular tests are more frequent in children infected for more than 7–10 days, which correlates to loss of infectivity and hence transmission. Our study shows that saliva molecular testing bypassing RNA extraction is an efficient method for monitoring SARS-CoV-2 infected children in age groups up to 10 years-old.

## Materials and methods

### Study design

A total of 49 adults and 85 children (aged 10 years and under) inpatients were enrolled in this study between 25 August 2020 and 20 June 2021. NP swab samples were collected and processed at the hospital. Saliva samples were collected from adults or children, within 24 or 48 h from NP swab collection, respectively. Both SARS-CoV-2 positive and negative individuals (by NP swab) were enrolled in this study. In adults, only symptomatic patients were enrolled, whereas, for children, patients were enrolled after admission to the hospital for COVID-19 symptoms or causes non-related to COVID-19 (other medical pathologies or surgeries).

### Saliva collection

At least 1 mL of saliva was collected with the help of a health care worker, after abstinence from food or water for at least 30 min. Participants were asked to pool saliva in the mouth and gently spit it into a sterile container without coughing or clearing their throats. For children under the age of 1 year, saliva was gently aspirated from the mouth with a suction tube. Samples were stored at 4˚C, sent to Instituto Gulbenkian de Ciência, and processed within 72 h from collection.

### SARS-CoV-2 detection

Saliva was treated with Proteinase K (645 μg/mL in 160 nM SDS) for 30 min at 50˚C, followed by heat inactivation for 10 min at 98˚C. Saliva specimens with high viscosity were diluted 1:2 in TBE 2x prior to Proteinase K treatment, as described in [18]. Samples were then used directly in the RT-qPCR reaction or after RNA extraction with the QIAamp Viral RNA Mini kit (Qiagen, 52906), according to the manufacturer's instructions; 200 μL saliva were extracted and eluted in 50 μL RNase-free water; 1 μL extracted RNA or unextracted saliva were used for RT-qPCR. A one-step assay (cDNA synthesis and amplification) was performed using iTaq Universal Probes One-Step Kit (BioRad, #12013250). A master mix was prepared for each set of primer-probe, CDC_N1, CDC_N2 and Hs_RPP30 using: 0.5 nM of each primer pair, 125 nM of probe, 1x iTaq Universal Probes reaction mix, 2.5% (V/V) iScript Reverse Transcriptase and 10% (V/V) RNA or unextracted saliva sample. Two positive controls were performed separately per experiment (SARS-CoV-2 and Human) for N1, N2, and RP primer-probe set using 2,000 synthetic copies of nucleocapsid region of the virus or 2,000 synthetic copies of the Human single copy RPP30 gene. One negative control was performed without template for the same three conditions (N1, N2 and RP). All the primers and probes (2019-nCoV RUO Kit, 500 rxn, #10006713) were purchased from Integrated DNA Technologies, as well as the positive virus detection control (2019-nCoV_N_Positive Control, #10006625) and positive human sample control (Hs_RPP30 Positive Control, #10006626). Reactions were performed in 384 wells plates (ThermoFisher, #TF-0384) in a QuantStudio 7 Flex system (Applied Biosciences), using the fast mode, consisting of a hold stage at 50˚C for 10 min, and 95˚C for 3 min, followed by 45 cycles of a PCR stage at 95˚C for 10 sec then 60˚C for 30 sec (FAM signal acquisition

step). Positive cases were considered when the two probes were amplified with a CT below 37. Negative detection was established as having no amplification or amplification of one probe above 37. Inconclusive results were considered, with only one probe being amplified with a CT less than 37. The limit of detection (LoD) of the saliva assay was performed by using 1:5 serial dilution of IDT synthetic copies (20,000 to 6.4) and then 1:2 dilutions (6.4–1.28 copies) of SARS-CoV-2 in fresh saliva samples (non-positives) to span the expected range of concentrations needed to determine LoD (S1 Fig). Briefly, a Probit regression analysis was performed using MedCal 20.027 software, and LoD for N1 and N2 were determined as 40.74 copies with a 95% coefficient interval (CI) of 17.54–738.16 and 20.39 copies with a 95% CI of 9.21–547.17, respectively.

## Infectious SARS-CoV-2 growth from saliva

Saliva samples were diluted 1:4 in virus growth medium (Dulbecco's Modified Eagle Medium [DMEM, Gibco, #21969035] supplemented with 2% fetal bovine serum [FBS, Gibco, #10500064], 2 mM L-glutamine [ThermoFisher, #25030024], 1% penicillin/streptomycin solution [Biowest, #L0022] and 2.5 µg/mL Amphotericin B [Gibco, #15290018]), vortexed, briefly centrifuged to collect debris and passed through a 40 µm filter. Two hundred µL of each sample were added in triplicate to Vero E6 cells (a kind gift from Rupert Beale, The Francis Crick Institute, UK) pre-seeded in 24 well plates with coverslips and centrifuged for 15 min at 3,500 g, 37˚C. After centrifugation, the inoculum was removed and 250 µL of fresh virus growth media were added to the cells (protocol adapted from [20]). As a positive control, cells were infected with SARS-CoV-2 viral stock (strain human/DEU/HH-1/2020 from the European Virus Archive Global) at a multiplicity of infection (MOI) of 0.01. Cells were inspected daily for cytopathic effect (CPE) and considered negative if no CPE was visible for 7 days. At 1- and 4-days post-infection, a replicate was fixed with 4% formaldehyde and processed for immunofluorescence, as described in [21]. Cells were stained with a rabbit polyclonal SARS-CoV-2 nucleocapsid antibody (1:1,000; ThermoFisher, #MA536270) and an anti-rabbit IgG Alexa-Fluor 488 secondary antibody (1:1,000; Invitrogen, #A-21206). Cell nuclei were stained with Hoechst 33342 (1 µg/mL, Invitrogen, #H3570). Single optical sections were imaged with a Leica SP5 confocal microscope.

## ELISA

The ELISA assay used to quantify saliva IgG, IgA, and IgM anti-full-length SARS-CoV-2 spike and its receptor-binding domain (RBD) was adapted from [22] as described in [23, 24], with few modifications described here. Briefly, high binding 96-well plates (Maxisorb) were coated with either RBD or spike as capture antigen at 0.5 µg/mL stored overnight at 4˚C. Plates were blocked with PBS supplemented with 2% BSA (PBS-BSA) for 1–4 hours at room temperature. Saliva samples were tested at 1 in 25 dilutions in PBS with 2% BSA and 0.1% Tween (PBS-BSA-T), in duplicate. For each isotype test plate, an IgG, IgM (GenScript [clone 2001]) or IgA (Absolute Antibody [clone 3022]) SARS-CoV-2 reactive monoclonal antibody was used to generate a concentration curve upon serial dilution and validate each plate assayed. Duplicate measurements of reference positive and negative sera samples were used at 1 in 50 dilutions in PBS-BSA-T to validate each plate assayed; negative sera were used to determine cutoff values.

Samples and monoclonal antibodies were incubated for 2 hours at 37˚C, then plates were washed 3x in PBS-T. Secondary antibody goat anti-Human Fc-HRP IgG, IgA or IgM (Abcam) diluted 1 in 25,000 in PBS-BSA-T was added to respective isotype plates and incubated for 1 hour at 37˚C to reveal bound IgG, IgA or IgM. Plates were then washed 3x in PBS-T and

incubated with 3,3′,5,5′-tetramethylbenzidine (BD OptEIA™, BD Biosciences) for 20 to 30 minutes at 37˚C. The reaction was stopped with sulphuric acid, and the colorimetric assay was read to provide optical density (OD) at 450 nm.

Reference negative sera were collected at least 3 years before the COVID-19 pandemic and used as pooled serum from 50 samples. Isotype-specific positive and negative thresholds were determined for each assay plate from the mean negative serum value plus two standard deviations (SD). Negative sera were obtained upon informed consent in the frame of the projects "Genetic susceptibility factors and immunologic protection in COVID-19" and "Genetic variance in Portuguese population: candidate genes in COVID-19", both approved by the IGC Ethics Committee (reference H004.2020 and H002.2020, respectively).

## Statistical analysis

The values of sensitivity, specificity, and accuracy were estimated using the results of the NP swab as the reference standard. 95% confidence intervals (CI) were calculated using the Wilson method, recommended for small sample sizes [25]. The analytical sensitivity of SARS-CoV-2 RNA detection between saliva and nasopharyngeal swabs from SARS-CoV-2 positive adults and children was compared using a Wilcoxon matched-pairs signed-rank test, in GraphPad Prism 9.1.2.

ELISA data were cleaned, and categorical variables were created where needed. Thresholds for antibody-positive responses were defined by the negative pool (mean + 2 SD). The correlation between antibody OD values and age was assessed using Pearson's Pairwise coefficients. Descriptive statistics were assessed as frequencies and percentages [n (%)] for categorical variables; crude associations were tested with Pearson's Chi2 tests. Significance level was set at <0.05. STATA (StataCorp LLC, USA, V16) was used for all analyses.

## Results

### Method validation in adults

We first validated our method with adult patients admitted to Hospital Fernando Fonseca with COVID-19, or without SARS-CoV-2 infection, as negative controls. Once SARS-CoV-2 infection was assessed by RT-qPCR on RNA extracted from an NP swab, a saliva sample was collected within the following 24 hours. Saliva specimens were pre-treated with proteinase K, RNA was extracted and subjected to RT-qPCR using the primer and probe sequences from the Center for Disease Control and Prevention [26]. Alternatively, saliva samples treated with proteinase K were used directly for the RT-qPCR reaction, without RNA extraction.

The overall concordances of saliva and NP swab were 98.0% (48/49) and 97.9% (46/47), with and without RNA extraction from saliva, respectively (Table 1). Of the patients with a positive NP swab, 100% were also positive in saliva, either with (36/36) or without (34/34) RNA extraction. The differences in CT values between saliva (with or without RNA extraction) and NP swab [median (IQR) of 22.0 (20.2–24.3)] were not statistically significant (Fig 1A), but CT values in saliva with [23.8 (20.7–27.5)] vs. without RNA extraction [24.9 (22.7–28.8)] are statistically different, with extraction of RNA consistently decreasing the CT values obtained. This result was already expected, since RNA extraction from samples results in a concentration of the RNA present and, at the same time, eliminates impurities that can interfere with the RT-qPCR reaction.

Next, we analyzed the stability of saliva samples stored for 3 days at 4˚C or 7 days at -20˚C, prior to processing (Fig 1B). We detected viral RNA in all saliva samples kept at 4˚C and -20˚C, regardless of RNA extraction, demonstrating they are stable under the conditions studied. Nevertheless, there is a modest but statistically significant increase in the CT values of

**Table 1. Performance of saliva in adults and children.** Summary of results obtained from parallel testing of NP swab and saliva samples with and without extraction of RNA.

| | | NP swab No. | Saliva No. | | | | | |
| --- | --- | --- | --- | --- | --- | --- | --- | --- |
| | | | with RNA extraction | | | without RNA extraction | | |
| | | | Positive | Negative | Total | Positive | Negative | Total |
| **Adults > 18y** | | | | | | | | |
| | Positive | | 36 | 0 | 36 | 34 | 0 | 34 |
| | Negative | | 1 | 12 | 13 | 1 | 12 | 13 |
| | Total | | 37 | 12 | 49 | 35 | 12 | 47 |
| | Sensitivity (95%CI) | | 100% (90.4%–100%) | | | 100% (89.8%–100%) | | |
| | Specificity (95%CI) | | 92.3% (66.7%–98.6%) | | | 92.3% (66.7%–98.6%) | | |
| | Accuracy (95%CI) | | 98.0% (89.3%–99.6%) | | | 97.9% (88.9%–99.6%) | | |
| **Children < 10y** | | | | | | | | |
| | Positive | | 39 | 7 | 46 | 36 | 8 | 44 |
| | Negative | | 0 | 39 | 39 | 0 | 39 | 39 |
| | Total | | 39 | 46 | 85 | 36 | 47 | 83 |
| | Sensitivity (95%CI) | | 84.8% (71.8%–92.4%) | | | 81.8% (68.0%–90.5%) | | |
| | Specificity (95%CI) | | 100% (91.0%–100%) | | | 100% (91.0%–100%) | | |
| | Accuracy (95%CI) | | 91.8% (84.0%–96.6%) | | | 90.4% (82.1%–95.0%) | | |
| **Children < 1y** | | | | | | | | |
| | Positive | | 20 | 3 | 23 | 19 | 3 | 22 |
| | Negative | | 0 | 5 | 5 | 0 | 5 | 5 |
| | Total | | 20 | 8 | 28 | 19 | 8 | 27 |
| | Sensitivity (95%CI) | | 87.0% (67.9%–95.5%) | | | 86.4% (66.7%–95.3%) | | |
| | Specificity (95%CI) | | 100% (56.6%–100%) | | | 100% (56.6%–100%) | | |
| | Accuracy (95%CI) | | 89.3% (72.8%–96.3%) | | | 88.9% (71.9%–96.1%) | | |

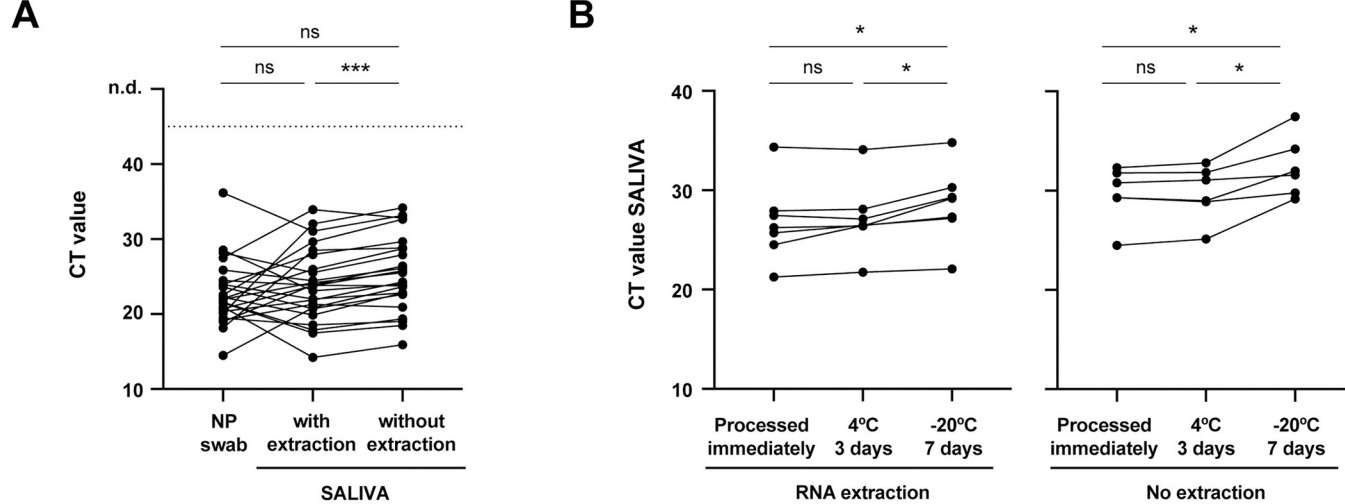

**Fig 1. SARS-CoV-2 RNA detection in NP swab and saliva samples from adult patients infected with SARS-CoV-2. (A)** Comparison of CT values from paired saliva (with and without RNA extraction) and NP swab specimens (N = 24). **(B)** Saliva stability at 4°C and -20°C: comparison of CT values from paired saliva samples [with RNA extraction (N = 7) and without RNA extraction (N = 6)] processed immediately, after 3 days at 4°C or 7 days at -20°C. Each line corresponds to a paired specimen. n.d., not detected. ns, not significant; *p<0.05, ***p<0.001, by Wilcoxon matched-pairs signed-rank test.

samples that were stored at -20˚C, suggesting that salivas with very high CT levels (close to 35) may become negative after storage at -20˚C.

## Performance of the method in pediatric patients

Eighty-five patients aged 10 years and under admitted to Hospital Dona Estefânia between August 2020 and June 2021 were included in our analysis. The clinical characteristics of the participants are shown in Table 2. Forty-six (54.1%) were positive for SARS-CoV-2 infection and, within these, 29 (63.0%) presented at least one COVID-19 symptom, the most common being fever (50.0%), cough (28.0%), and coryza (28.0%). The remaining 17 children (37.0%) were diagnosed on routine tests prior to hospital admission for causes non-related to COVID-19. Twenty-eight (32.9%) participants were under the age of 1 and the mean (SD) age was 3.8 (3.4) years.

Upon hospital admission, SARS-CoV-2 infection was assessed by RT-qPCR quantification of RNA load in NP swab samples. Saliva samples were collected within the following 48 hours

**Table 2. Characteristics and reported signs and symptoms of SARS-CoV-2-positive and negative children included in this study.**

| Characteristic | | No. (%) | | |
|---|---|---|---|---|
| | | Total sample | Negative | Positive |
| **Total** | | 85 | 39 (45.9) | 46 (54.1) |
| **Sex** | | | | |
| | Female | 39 (45.9) | 20 (51.3) | 19 (41.3) |
| | Male | 46 (54.1) | 19 (48.7) | 27 (58.7) |
| **Age (y)** | | | | |
| | <1 | 28 (32.9) | 5 (12.8) | 23 (50) |
| | 1–5 | 25 (29.4) | 15 (38.5) | 10 (21.7) |
| | 6–10 | 32 (37.7) | 19 (48.7) | 13 (28.3) |
| **COVID-19 sign or symptom** | | | | |
| | None | 17 (37.0) | NA | 17 (37.0) |
| | Fever | 23 (50.0) | NA | 23 (50.0) |
| | Cough | 13 (28.3) | NA | 13 (28.3) |
| | Dyspnea | 5 (10.9) | NA | 5 (10.9) |
| | Coryza | 13 (28.3) | NA | 13 (28.3) |
| | Odynophagia | 4 (8.7) | NA | 4 (8.7) |
| | Headache | 1 (2.2) | NA | 1 (2.2) |
| | Abdominal pain | 1 (2.2) | NA | 1 (2.2) |
| | Nausea/Vomit | 3 (6.5) | NA | 3 (6.5) |
| | Diarrhea | 3 (6.5) | NA | 3 (6.5) |
| **Concurrent conditions** | | | | |
| | 0 | 11 (12.9) | 1 (2.6) | 10 (21.7) |
| | 1 | 69 (81.2) | 35 (89.7) | 34 (73.9) |
| | >1 | 5 (5.9) | 3 (7.7) | 2 (4.3) |
| | Another Infection | 12 (14.1) | 1 (2.6) | 11 (23.9) |
| | Cardiovascular disease | 5 (5.9) | 0 | 5 (10.9) |
| | Urinary tract disease | 13 (15.3) | 6 (15.4) | 7 (15.2) |
| | Digestive tract disease | 10 (11.8) | 5 (12.8) | 5 (10.9) |
| | Another respiratory disease | 5 (5.9) | 0 | 5 (10.9) |
| | Oral surgery | 4 (4.7) | 4 (10.2) | 0 |
| | Facial congenital anomalies | 3 (3.5) | 3 (7.7) | 0 |

and processed as described above. The overall concordances of saliva and NP swab were 91.8% (78/85) with RNA extraction and 90.4% (75/83) without RNA extraction from saliva (Table 1). Of the children with a positive NP swab, 84.8% were positive in saliva after RNA extraction (39/46) and 81.1% (36/44) were positive in saliva without RNA extraction. The specificity of the method was 100%, with the 39 patients negative by NP swab, also negative in saliva, with and without RNA extraction. When restricting the analysis to children younger than 1-year-old, the sensitivity of the method increases to 87.0% (25/28) with RNA extraction and 86.4% (19/27) without RNA extraction from saliva (Table 1).

The CT values in the saliva of children were significantly different from those in NP swab (Fig 2A), with a mean CT (SD) of 22.9 (6.2) in NP swab versus 26.1 (5.1) in saliva with RNA extraction and 27.9 (4.7) without extraction. It is important to notice that there was a time interval between NP swab and saliva collection that could go up to 48 h and, therefore, these differences may not reflect a lower sensitivity of our method but rather a decrease in the patient viral load. Importantly, there was no correlation between detection of SARS-CoV-2 in saliva and the CT value of the NP swab, or the age of the patient (Fig 2B).

We also analyzed the stability of saliva samples from children when stored for 3 days at 4˚C or 7 days at -20˚C prior to processing (Fig 2C). As for adults, saliva samples from children were stable under the conditions tested, since viral RNA was detectable in all samples, irrespective of whether RNA extraction was performed. Furthermore, we did not observe the increase in CT values after storage at -20˚C that was registered for adult patients. One possible explanation for this difference is that adults and children are known to have different saliva composition, which may alter the stability of the viral RNA during the freezing/thawing processes.

Finally, we tested saliva samples on a commercial lateral-flow test (COVID19 antigen rapid test, ALL TEST, #ICOV-502). Saliva samples that tested positive in our molecular assay with CT values<26 also tested positive in the chosen lateral-flow assay (S2 Fig). Of note, two other commercial lateral-flow tests performed poorly in our hands. These data are consistent with what was observed for NP swabs [16] and may have direct implications for monitoring schools.

## Viral growth from saliva samples

To address if we could detect infectious SARS-CoV-2 viruses in children saliva and correlate viral replication with their CT value, saliva samples were cultured in Vero cells, inspected daily for CPE, and analyzed by immunofluorescence using an antibody against SARS-CoV-2 nucleoprotein (Fig 2D and 2E). Importantly, we recovered infectious viruses from all saliva samples with CT values ≤25.6, whereas in saliva samples with higher CT values or negative, no viral replication was detected. These results suggest that children with CT values in saliva equal or higher than 26 may not be able to shed infectious viruses.

## Antibodies against SARS-CoV-2 in saliva

To address the levels of anti-SARS-CoV-2 IgG, IgA and IgM antibodies in the saliva of adults and children, we performed ELISA assays against full-length spike and spike's RBD (Fig 2F and Table 3). In adults, all saliva samples (n = 21) tested negative for IgG reactivities. In contrast, reactive IgA was detected in 16 (76.2%) adults, regardless of NP swab and saliva results, suggesting that the detection of IgA antibodies does not correlate with infection. This discrepancy could, however, be explained by the presence of secreted IgAs (sIgAs) that recognize glycosylation patterns existent in spike [27, 28]. Only one (4.8%) participant was positive for reactive IgM.

Of the 73 saliva samples from children tested by ELISA, 3 (3.1%) were positive for specific IgG. Reactive IgA and IgM were detected in 44 (60.3%) and 5 (11.1%) participants,

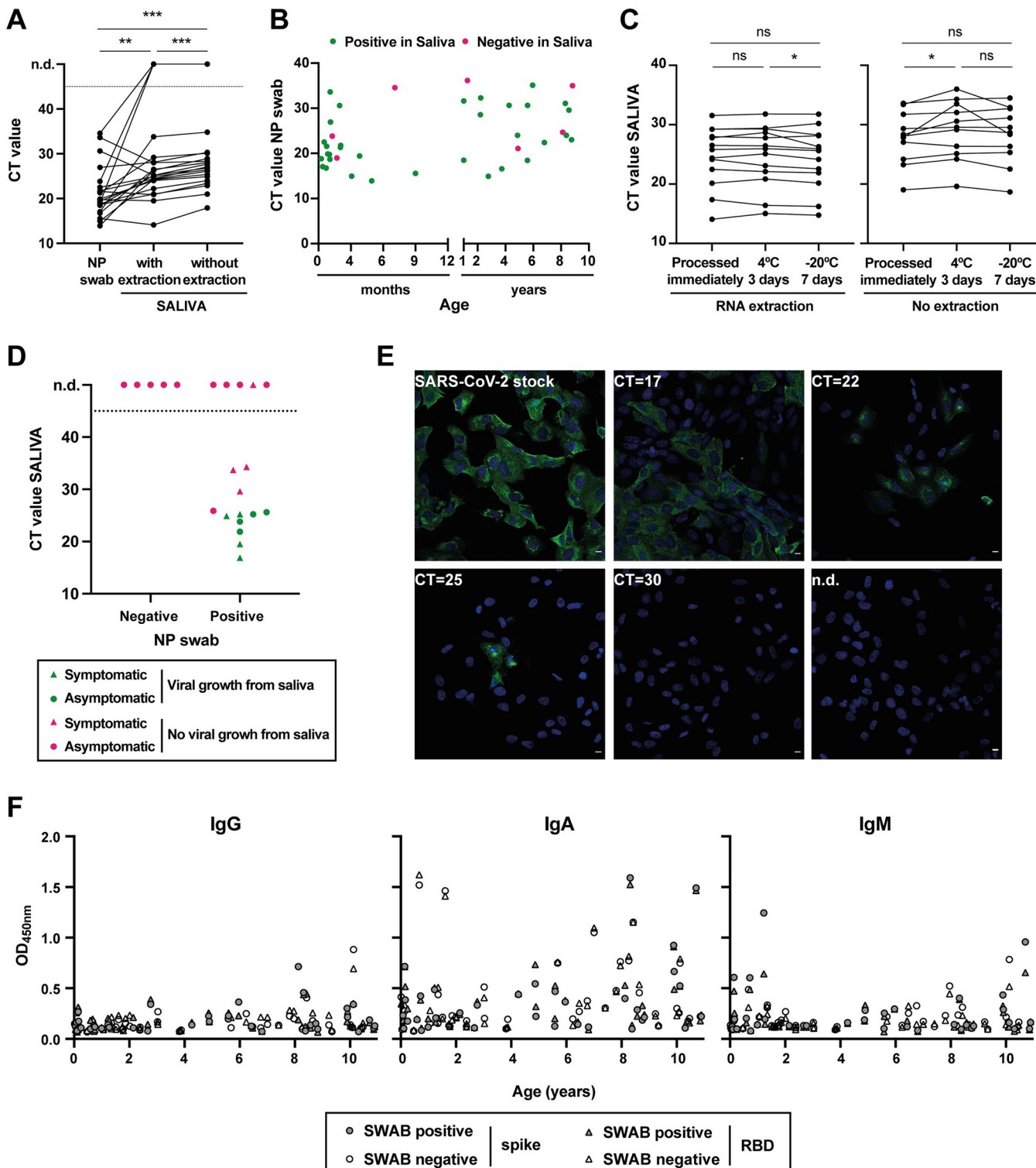

**Fig 2. Detection of SARS-CoV-2 infection in NP swab and saliva samples from children. (A)** Comparison of CT values from paired saliva (with and without RNA extraction) and NP swab specimens from children aged 10 years and under, positive for SARS-CoV-2 infection (N = 40). Each line corresponds to a paired specimen. n.d., not detected; **p<0.01, ***p<0.001, by Wilcoxon matched-pairs signed-rank test. **(B)** Graphical representation of CT values in NP swabs from infected children vs. age. Positive in saliva, green dots; negative in saliva, magenta dots. **(C)** Saliva stability at 4°C and -20°C: comparison of CT values from paired saliva samples [with (N = 13) and without RNA extraction (N = 11)] processed immediately, after 3 days at 4°C or 7 days at -20°C. ns, not significant; *p<0.05, by Wilcoxon matched-pairs signed-rank test. **(D, E)** Infectious SARS-CoV-2 growth from saliva samples. Vero E6 cells were inoculated

with saliva samples from children and inspected daily for the presence of a cytopathic effect. As a positive control, cells were infected with SARS-CoV-2 viral stock at an MOI of 0.01. **(D)** Graphical representation of CT values in saliva vs. NP swab result, with dots representing symptomatic children, triangles asymptomatic patients, green color indicating samples where viral replication was detected, and magenta samples without viral growth. n.d., not detected (N = 22). **(E)** At 24h post-infection, cells were fixed with 4%formaldehyde, permeabilized with 0.2% Triton X100, and stained with SARS-CoV-2 Nucleocapsid antibody (green). Cell nuclei were stained with Hoechst (blue). White bar = 10μm. Images were acquired with a Leica SP5 confocal microscope. Representative images from Vero cells infected with SARS-CoV-2 viral stock or inoculated with saliva specimens with CT values of 17, 22, 25, and 30, or saliva with non-detected SARS-CoV-2 (n.d.). **(F)** Levels of IgG, IgA, and IgM against full-length spike and spike's RBD measured in the saliva of children aged 10 years and under by ELISA.

respectively. Four participants with discrepant NP vs. saliva RT-qPCR data (NP positive, saliva negative, n = 12) presented IgM, suggesting that the corresponding samples were collected after the first immune response, estimated to be 7–10 days post-infection, and therefore after the peak of viral loads.

For children up to 10 years, IgG OD responses increased significantly ($p<0.05$) with age for spike, while IgA OD responses to spike and RBD presented a marginally significant ($p<0.1$) increase with age (Fig 2F).

## Discussion

The infection of SARS-CoV-2 in children remains under-diagnosed and poorly understood. In particular, the role of children in viral transmission remains unclear, especially with the emerging variants that increase viral transmission, such as the delta and the recently detected omicron variants. In our work, we corroborate findings that saliva molecular testing by methods similar

**Table 3. Patients with antibodies in saliva against full-length spike or RBD, divided by age category and by RT-qPCR result on swab and saliva.**

| Group | Total sample No. | Positive No. (%) | | | | | |
|---|---|---|---|---|---|---|---|
| | | Spike | | | RBD | | |
| | | IgG | IgA | IgM | IgG | IgA | IgM |
| **Adults > 18y** | **21** | **0** | **15 (71.4)** | **1 (4.8)** | **0** | **16 (76.2)** | **1 (4.8)** |
| Swab-/saliva- | 5 | 0 | 3 (60.0) | 0 | 0 | 3 (60.0) | 0 |
| Swab-/saliva+ | 0 | 0 | 0 | 0 | 0 | 0 | 0 |
| Swab+/saliva- | 0 | 0 | 0 | 0 | 0 | 0 | 0 |
| Swab+/saliva+ | 16 | 0 | 12 (75.0) | 1(6.3) | 0 | 13 (81.3) | 1 (6.2) |
| **Children ≤ 10y** | **73** | **3 (4.1)** | **44 (60.3)** | **5 (11.1)** | **1 (1.37)** | **39 (53.4)** | **4 (5.5)** |
| Swab-/saliva- | 37 | 2 (5.4) | 21 (56.8) | 1(2.70) | 0 | 20 (54.0) | 0 |
| Swab-/saliva+ | 0 | 0 | 0 | 0 | 0 | 0 | 0 |
| Swab+/saliva- | 12 | 1 (8.3) | 9(75.0) | 4 (33.3)** | 0 | 7 (58.3) | 3(25.0)* |
| Swab+/saliva+ | 24 | 0 | 14 (58.3) | 0 | 1 (4.17) | 12 (50.0) | 1 (4.17) |
| **Total (Adults+Children)** | **94** | **3 (3.2)** | **59 (62.8)** | **6 (6.4)** | **1 (1.06)** | **55 (58.6)** | **5 (5.3)** |
| Swab-/saliva- | 42 | 2 (4.8) | 24 (57.1) | 1 (2.4) | 0 | 23 (54.8) | 0 |
| Swab-/saliva+ | 0 | 0 | 0 | 0 | 0 | 0 | 0 |
| Swab+/saliva- | 12 | 1 (8.3)* | 9 (75.0) | 4 (33.3)** | 0 | 7 (58.3) | 3 (25.0)** |
| Swab+/saliva+ | 40 | 0 | 26 (65.0) | 1 (2.50) | 1 (2.50) | 25 (62.5) | 2(5.00) |
| OD$_{450nm}$ Mean (SD) | 94 | 0.17 (0.13) | 0.41 (0.35) | 0.21 (0.18) | 0.17 (0.1) | 0.43 (0.37) | 0.19 (0.12) |
| OD$_{450nm}$ Min-Max | 94 | 0.07–0.88 | 0.08–1.59 | 0.07–1.24 | 0.07–0.69 | 0.09–1.62 | 0.07–0.66 |

+, positive; -, negative

*$p<0.1$

**$p<0.05$ by Pearson's Chi2 test. The tests were applied to compare the number of positives of the same antibody between the four groups: swab and saliva negative (swab-/saliva-); swab negative and saliva positive (swab-/saliva+); swab positive and saliva negative (swab+/saliva-); and swab and saliva positive (swab+/saliva+).

to those described previously [15] efficiently detects infected children, even in children unable to donate saliva (Fig 2A), from whom saliva was gently aspirated. Importantly, in children, the sensitivity, specificity, and accuracy of saliva-RT-qPCR tests compared to NP swab-RT-qPCR were respectively 84.8% (71.8%–92.4%), 100% (91.0%–100%), and 91.8% (84.0%–96.6%) with RNA extraction and 81.8% (68.0%–90.5%), 100% (91.0%–100%), and 90.4% (82.1%–95.0%) without RNA extraction. Hence, we show that methods bypassing RNA extraction have a sensitivity of 97.3% (36/37) compared with assays extracting RNA, and 100% (36/36) sensitivity considering CTs up to 36 with RNA extraction. Interestingly, evidence is accumulating that the omicron variant has a different tropism, infecting preferentially the upper respiratory tract [29–32]. In addition, it has been reported that saliva-molecular testing results in a higher positive percent agreement when compared to NP swab molecular detection in the omicron variant than in the delta variant [33]. These data suggest that saliva may be the preferred sample used for future monitoring. Interestingly, in our study, the observed discrepancies between saliva and NP swab-based molecular tests were found in children with specific IgM responses, which could suggest lower viral loads and prolonged infection. Interestingly, our study detected a high proportion of uninfected children with IgA antibodies in saliva. Antibodies in saliva may be acquired from the blood via the gingival crevicular fluid [34]. However, there is a portion of IgAs that are produced locally in mucosal tissues, such as salivary glands, including secretory IgA (sIgA) [27]. While some sIgA antibodies undergo affinity maturation, others recognize specific glycosylation patterns [28] and could justify the high levels detected in negative patients. This cross-reactive response has been reported by others in the mucosal tissue [35, 36] and is considered an important property of sIgAs for protecting the mucosal tissue against infections.

In addition, our study shows that SARS-CoV-2 infectious viruses may be rescued from saliva samples with CT values lower than 26 (Fig 2D and 2E) and previous studies found that it is possible to sequence viruses directly from saliva, meaning that it is a suitable biological sample not only to identify infected people, but also to assess the infectivity status and the viral variant [37]. Data suggest that high viral loads are prone to result in viral transmission [20]. In our study, SARS-CoV-2 viral growth was not associated with symptoms in children, but the sample size used was too low to draw conclusions regarding how symptoms relate to viral loads. In fact, it has been shown that asymptomatic children have significantly lower viral loads than symptomatic children [7]. Overall, it is well-established that infected children are more likely to remain asymptomatic or have milder disease than adults; children are rarely hospitalized and rarely have fatal outcomes [1, 6, 9, 10, 38]. However, data accumulated from 22 centers throughout South Korea has shown that 58% of symptomatic infected children experienced symptoms 3 days (median, in a range of 1–28 days) before SARS-CoV-2-positive diagnostic and that, despite having a detectable virus load, pre-symptomatic children remained symptom-free for 2.5 days (median, in a range of 1–25 days) [6]. For these reasons, the vast majority of SARS-CoV-2 infected children will continue to be missed by a symptom-based testing approach [6]. With the unprecedented effort of vaccinating the world population above 12-years-old, it is critical to understand how children below 12-years-old, the majority still unvaccinated, contribute to viral circulation within their community. To this need, it is essential to implement efficient, easy, non-invasive, cheap methods for accurately identifying and tracking infected children, for which our method offers a solution. It is also critical to establish if children are a potential source for emerging novel SARS-CoV-2 variants, and saliva is suitable for collecting SARS-CoV-2 for genotyping [13–15] and detecting antibodies specific to spike and RBD as we show here (Fig 2F and Table 3). How infection in children will impact breakthrough infections upon vaccination, which has been demonstrated as possible [39], remains to be elucidated and is of the utmost importance.

## Limitations

This cross-sectional study has one key limitation. There was an interval of up to 48 hours between SARS-CoV-2 infection diagnostic in children by NP swab and the collection of a saliva sample. Given this time interval, it remains unclear whether the saliva molecular testing has a lower sensitivity than the NP swab test, or whether there was a real decrease in the patient viral load between the two samples collection. In support of the former scenario, other studies conducted in adults reported a small detection decrease in saliva relative to NP swabs [15, 40–44], which may not be true for the new variants. In fact, recent studies with the new Omicron variant suggest that this variant displays improved replication in the upper respiratory tract, with higher viral shedding in saliva relative to nasopharyngeal mucosa [33]. Despite the temporal limitation, our saliva testing in children up to 10-years-old indicates a high sensitivity, specificity, and accuracy with and without RNA extraction, showing that the method is suitable for detecting infected children. An important note is that children were admitted to hospital for many reasons other than COVID-19 related symptoms and hence this study is suitable to draw conclusions for children regardless of their symptoms. A second limitation of this study is that the number of samples from which we rescued infective viral particles in cultured cells does not allow a suitable statistical analysis to relate viral loads with symptoms and is not a formal demonstration of transmission from children. A third caveat is that the ELISA assay, despite being specific for sera and collected from infected people [23], was not established for saliva. In fact, we did not calibrate the ELISA with pre-pandemic saliva, and as children were not re-analyzed posteriorly for antibody development, there is a lack of saliva positive and negative controls, which also limits the conclusions we may draw from the data.

## Conclusions

Saliva-molecular testing is suitable in children aged 10 years and under, including infants aged <1 year, even bypassing RNA extraction methods. Importantly, the detected viral RNA levels were significantly above the infectivity threshold in several samples. Overall, our study provides a method highly suitable to identify children positive for SARS-CoV-2. It could be used for the surveillance of kindergartens and schools, and also as the first step in genotyping efforts to monitor known variants and spot novel ones when coupled to CRISPR-based methodologies [45].

## Supporting information

**S1 Fig. Limit of detection (LoD) of the saliva RT-qPCR assay was evaluated using 9 serial dilutions of synthetic SARS-CoV-2 sequences corresponding to portions of N gene, made in fresh non-positive salivas, each with at least 5 replicates.** Probit analysis (plot of fitted model) was performed using the MedCalc software to determine the LoD by fitting template copies (on a logarithmic scale) against the cumulative fractions of positive PCR observations (blue line), and to calculate the lower and upper 95% confidence intervals (CI) (red dashed line). LoD for N1 probe was defined at 40.74 copies with a 95% CI of 17.54–738.16 (**A**), and at 20.39 copies for N2 probe with a 95% CI of 9.21–547.17 (**B**). Detection linearity and correlation between CTs and copy number was observed between 20 and 20000 copies (data not shown).
(TIF)

**S2 Fig. COVID-19 rapid antigen test detects SARS-CoV-2 in saliva samples positive by RT-qPCR and with CT values up to 26.** Graphical representation of CT values in saliva after RNA extraction vs. rapid antigen test result, with dots representing symptomatic children,

triangles asymptomatic patients. n.d., not-detected. The COVID-19 antigen rapid test from ALL TEST (ref. ICOV-502) was used accordingly to the manufacturer's instructions, with the exception that instead of NP swab, saliva samples diluted 1:2 in extraction buffer were used. (TIF)

**S1 File. Data for Tables 1–3 & Figs 1 and 2.**
(XLSX)

## Acknowledgments

We would like to thank the technical support of IGC's Advanced Imaging Facility (AIF-UIC).

## Author Contributions

**Conceptualization:** Marta Alenquer, Maria João Amorim.

**Data curation:** Marta Alenquer, Onome Akpogheneta, Filipe Ferreira.

**Formal analysis:** Marta Alenquer, Onome Akpogheneta, Filipe Ferreira, Jocelyne Demengeot, Maria João Amorim.

**Funding acquisition:** Vasco M. Barreto, Jocelyne Demengeot, Maria João Amorim.

**Investigation:** Marta Alenquer, Tiago Milheiro Silva, Onome Akpogheneta, Filipe Ferreira.

**Methodology:** Marta Alenquer, Tiago Milheiro Silva, Onome Akpogheneta, Filipe Ferreira, Sílvia Vale-Costa, Mónica Medina-Lopes, Frederico Batista, Cathy Paulino, João Costa, João Sobral, Maria Diniz-da-Costa, Susana Ladeiro, Rita Corte-Real, Ricardo B. Leite.

**Project administration:** Maria João Amorim.

**Resources:** Tiago Milheiro Silva, Frederico Batista, Ana Margarida Garcia, Rita Corte-Real, José Delgado Alves, Ricardo B. Leite, Maria João Rocha Brito.

**Software:** Ricardo B. Leite.

**Supervision:** Maria João Amorim.

**Validation:** Marta Alenquer, Onome Akpogheneta, Ricardo B. Leite.

**Visualization:** Marta Alenquer, Onome Akpogheneta, Ricardo B. Leite.

**Writing – original draft:** Marta Alenquer, Onome Akpogheneta, Maria João Amorim.

**Writing – review & editing:** Marta Alenquer, Tiago Milheiro Silva, Onome Akpogheneta, Filipe Ferreira, Sílvia Vale-Costa, Mónica Medina-Lopes, Vasco M. Barreto, Cathy Paulino, João Costa, João Sobral, Maria Diniz-da-Costa, Susana Ladeiro, Ricardo B. Leite, Jocelyne Demengeot, Maria João Rocha Brito, Maria João Amorim.

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
