## [Decision Letter · Decision Letter 0]

11 Feb 2022

PONE-D-22-00630Saliva molecular testing bypassing RNA extraction is suitable for monitoring and diagnosing SARS-CoV-2 infection in childrenPLOS ONE

Dear Dr. Amorim,

Thank you for submitting your manuscript to PLOS ONE. After careful consideration, we feel that it has merit but does not fully meet PLOS ONE’s publication criteria as it currently stands. Therefore, we invite you to submit a revised version of the manuscript that addresses the points raised during the review process.

We look forward to receiving your revised manuscript.

Kind regards,

Ruslan Kalendar

Academic Editor

PLOS ONE

Journal Requirements:

"We would like to thank the technical support of IGC's Advanced Imaging Facility (AIF-UIC), which is supported by the national Portuguese funding ref# PPBI-POCI-01-0145-FEDER-022122, co-financed by Lisboa Regional Operational Programme (Lisboa 2020), under the Portugal 2020 Partnership Agreement, through the European Regional Development Fund (FEDER) and Fundação para a Ciência e a Tecnologia (FCT; Portugal)."

"This work was funded by the Fundação para a Ciência and Tecnologia (FCT, Portugal) under RESEARCH4COVID 19 call with reference 283_596885654 and co-funded by ANI under INOV4COVID (Funding to V.M.B.). M.J.A. is funded by the FCT (CEECIND/02373/2020). M.A. is funded by a Junior Researcher working contract from FCT and Instituto Gulbenkian de Ciência (IGC, Portugal). This work benefited from COVID19 emergency funds 2020 from Calouste Gulbenkian Foundation and from Oeiras city council. 

Reviewers' comments:

Reviewer's Responses to Questions

**Comments to the Author**

1. Is the manuscript technically sound, and do the data support the conclusions?

Reviewer #1: Yes

Reviewer #2: Yes

2. Has the statistical analysis been performed appropriately and rigorously? 

Reviewer #1: Yes

Reviewer #2: Yes

3. Have the authors made all data underlying the findings in their manuscript fully available?

Reviewer #1: No

Reviewer #2: Yes

4. Is the manuscript presented in an intelligible fashion and written in standard English?

Reviewer #1: Yes

Reviewer #2: Yes

5. Review Comments to the Author

Reviewer #1: 

This work investigates if saliva is an effective sample for detecting SARS-CoV-2 RNA and antibodies, especially in children. They selected a group of 85 children under 10 years old, and found that 63% have at least one COVID-19 symptom, 54.1% are positive cases. From where, they concluded that Saliva is effective for children under 10. This topic is helpful for recent covid-19 omicron variant detection. The writing style is scientific and easy to follow. However, the topic is not new, since it has been talked about for a while and many other research groups have done similar works in this area. Besides, there are several details I would like authors to address.

Given these unsolved questions above, a major revision of this manuscript is required.

Major concerns:

The adults over 18 have been chosen if they have at least one symptom, while for children under 10 there is not a such restriction. Why did author use those two data?

Minor concern:

Figure resolution. Not sure if pictures shown are from their original resolution, but the resolution in my end is really low. I barely can see details clearly. Please double check the submission of figures. Some other concerns about figures:

• Figure2 involves four different colors in panel B and D, but cannt see clear differences between blue green and pink red (I guessed the color). Maybe due to the resolution, but I highly recommend the authors to change four different colors or shapes with high contrasts.

• Figure S2 has a x-scale of 10 – 10000. Whats the reason for doing so? Did you did any log transformation? Please clarify.

Reviewer #2: 

Alenquer et al. evaluated an alternative method respect to the nasopharyngeal swab for the detection and quantification of SARS-CoV-2. In detail, they investigated whether saliva is an effective sample for detecting RNA in children, and assessed a possible association between viral RNA load and infectivity. Focusing on the detection of SARS-CoV-2 RNA, the authors devised two methods: one involving RNA extraction and the other not. They demonstrated that the saliva molecular test bypassing RNA extraction is a valid and useful method for surveillance.

The manuscript is interesting, even if some statements need to be clarified and proven.

These are my main concerns:

- Lines 133-135: “The limit of detection (LOD) of the saliva assay was performed by serial dilution of IDT synthetic copies (20,000 to 20) of SARS-CoV-2 in fresh saliva samples (non-positives) (S1 Fig)”. In detail, how was the detection limit of the assay evaluated? Which kind of tests were applied to calculate this limit? Have the authors performed a probit regression analysis? In addition, the coefficient of variation and the lower limit of blank (LoB) should be evaluated to better define the accuracy and limit of detection.

- Table 1- Adults: in the table it is reported that a negative swab sample was positive in saliva both with and without extraction. Were these assays performed on the same sample? How can the authors explain these different results? Have further swabs been performed on the patient to see if it is a really false positive in saliva?

- Table 1- children <1y and <10y: Did the samples that tested positive to swab but negative to saliva belong to the same patient? For these samples is there a correlation between the result in saliva and the CT value in the swab?

- Lines 211-214: “The differences in CT values between saliva (with or without RNA extraction) and NP swab were not statistically significant (Fig 1A), but CT values in saliva with vs. without RNA extraction are statistically different, with extraction of RNA consistently decreasing the CT values obtained.” To better understand the data, it would be useful to report the median values (IQR) of the Ct values obtained by the 3 detection methods.

- How can the authors explain the fact that samples with RNA extraction had lower labours than those without extraction?

- Figure 1: Please report sample size

- Lines 228-230: “Next, we analyzed the stability of saliva samples stored for 3 days at 4ºC or 7 days at -20ºC, prior to processing (Fig 1B). We detected viral RNA in all saliva samples kept at 4ºC and -20ºC, regardless of RNA extraction, demonstrating they are stable under the conditions studied.” The detection of the virus has not changed, but if you look at the figure 1B, a clear increase in CT in samples not extracted and stored at -20 for 7 days can be appreciated. Have the authors defined if this increase is statistically significant?

Moreover, this may affect the outcome of the result. For example, if the sample processed immediately had 35 Ct there is a high risk that it will be negative after a -20 storage 7 days long.

- Table 2: Please report the % of Concurrent conditions.

- Lines 257-260: “It is important to notice that there was a time interval between NP swab and saliva collection that could go up to 48 h and, therefore, these differences may not reflect a lower sensitivity of our method but rather a decrease in the patient viral load.” Stratifying the samples as: collected within 24 hours of TNF and collected 24-48 hours later, did you see differences in terms of CT values? And if so, did the former have values more similar to those obtained in TNF?

- Figure 1 e 2: “Saliva stability at 4ºC and -20ºC: comparison of CT values from paired saliva samples (with and without RNA extraction) processed immediately, after 3 days at 4ºC or 7 days at -20ºC.” Non-extracted samples do not seem similar between adults and children. Is there an explanation for this?

- Table 3: please define the populations used to calculate p-values.

- No symptoms concerning the lower respiratory tract (pneumonia, bronchiolitis) were described in the sampled population. This does not allow to define the specificity and sensitivity of the salivary method in detecting SARS-CoV-2 in subjects developing lower respiratory tract infections respect to those developing upper respiratory tract symptoms only. This limitation should be mentioned in the discussion section.

6. PLOS authors have the option to publish the peer review history of their article (what does this mean?). If published, this will include your full peer review and any attached files.

Reviewer #1: No

Reviewer #2: No

---

## [Author Response · Author response to Decision Letter 0]

28 Mar 2022

Authors response to reviewers

 -> We thank the reviewers for taking the time to read our manuscript and for providing valuable suggestions. We altered the manuscript to comply with both reviewers. We split the comments per reviewer and answered each reviewer’s question separately. Answers are in blue in sentences that start with an arrow (->), for making it easy to track.

Reviewer #1: 

This work investigates if saliva is an effective sample for detecting SARS-CoV-2 RNA and antibodies, especially in children. They selected a group of 85 children under 10 years old, and found that 63% have at least one COVID-19 symptom, 54.1% are positive cases. From where, they concluded that Saliva is effective for children under 10. This topic is helpful for recent covid-19 omicron variant detection. The writing style is scientific and easy to follow. However, the topic is not new, since it has been talked about for a while and many other research groups have done similar works in this area. Besides, there are several details I would like authors to address.

Given these unsolved questions above, a major revision of this manuscript is required.

->We thank this reviewer for the comments on our work. We complied with the suggestions and answered the questions below each comment. We submitted our work in July of last year to a journal that has only replied in January 2022, and that is the reason why our work is not that novel anymore. Nevertheless, given that children are rarely assessed, we still think that the work we developed is valid and novel, especially since we combined PCR molecular detection with antibody type response.

Major concerns:

The adults over 18 have been chosen if they have at least one symptom, while for children under 10 there is not a such restriction. Why did author use those two data?

-> The aim of our study was to address the suitability of saliva as an alternative sample for SARS-CoV-2 detection in children. We used samples from adults over 18 only as a first approach to establish the method that in our hands would then be used to screen children, regardless of infection status and symptoms. Therefore, we only assessed symptomatic adults to be sure that our method could work. 

Minor concern:

Figure resolution. Not sure if pictures shown are from their original resolution, but the resolution in my end is really low. I barely can see details clearly. Please double check the submission of figures. 

->We do not know why the reviewer cannot see our pictures with high resolution. Our figures were made accordingly to submission guidelines and uploaded with the recommended resolution of 300dpi. In our hands, it helps when we download the figures to our own computer. 

Some other concerns about figures:

• Figure2 involves four different colors in panel B and D, but cannt see clear differences between blue green and pink red (I guessed the color). Maybe due to the resolution, but I highly recommend the authors to change four different colors or shapes with high contrasts.

->Figure 2B and 2D have only 2 colours: green and red. To make sure that the colours are clearly distinguishable, we have changed them to green and magenta

• Figure S2 has a x-scale of 10 – 10000. Whats the reason for doing so? Did you did any log transformation? Please clarify.

->Figure S2 does not have an x-scale. We believe that the Reviewer is referring to Figure S1, in which the x axis is in log scale. To comply with reviewer 2 request, we have changed this figure to show the probit analysis performed to calculate the limit of detection of the assay. The x axis on the new plots are also in log scale, and this is clearly stated in the figure legend.

Reviewer 2:

Alenquer et al. evaluated an alternative method respect to the nasopharyngeal swab for the detection and quantification of SARS-CoV-2. In detail, they investigated whether saliva is an effective sample for detecting RNA in children, and assessed a possible association between viral RNA load and infectivity. Focusing on the detection of SARS-CoV-2 RNA, the authors devised two methods: one involving RNA extraction and the other not. They demonstrated that the saliva molecular test bypassing RNA extraction is a valid and useful method for surveillance.

The manuscript is interesting, even if some statements need to be clarified and proven.

->We thank this reviewer for the constructive comments on our work. We have changed the manuscript taking all the concerns of this reviewer into consideration.

These are my main concerns:

- Lines 133-135: “The limit of detection (LOD) of the saliva assay was performed by serial dilution of IDT synthetic copies (20,000 to 20) of SARS-CoV-2 in fresh saliva samples (non-positives) (S1 Fig)”. In detail, how was the detection limit of the assay evaluated? Which kind of tests were applied to calculate this limit? Have the authors performed a probit regression analysis? In addition, the coefficient of variation and the lower limit of blank (LoB) should be evaluated to better define the accuracy and limit of detection.

->We have expanded our analysis of the limit of detection to meet the reviewer’s requests and changed the text and figure S1 accordingly. The limit of detection (LoD) of the saliva assay was performed by using 1:5 serial dilution of IDT synthetic copies (20,000 to 6.4) and then 1:2 dilutions (6.4-1.28 copies) of SARS-CoV-2 in fresh saliva samples (non-positives) to span the expected range of concentrations needed to determine LoD. Briefly, a Probit regression analysis was performed using MedCal 20.027 software, and the LoDs for N1 and N2 were determined as 40.74 copies with a 95% coefficient interval (CI) of 17.54-738.16 and 20.39 copies with a 95% CI of 9.21-547.17, respectively.

- Table 1- Adults: in the table it is reported that a negative swab sample was positive in saliva both with and without extraction. Were these assays performed on the same sample? How can the authors explain these different results? Have further swabs been performed on the patient to see if it is a really false positive in saliva?

->The assays with and without RNA extraction were always performed on the same saliva sample. We have not performed further swabs on this patient, and we have no way of knowing for sure whether this result corresponds to a false positive in saliva or a false negative on the swab. However, since the Ct values on saliva were low (25.2 and 27.7, with and without extraction, respectively), we favour the second hypothesis. 

- Table 1- children <1y and <10y: Did the samples that tested positive to swab but negative to saliva belong to the same patient? For these samples is there a correlation between the result in saliva and the CT value in the swab?

->Yes, the samples swab positive and saliva negative belong to the same patients. As can be seen in figure 2B, there is no correlation between the result in saliva and the CT value in the swab. 

- Lines 211-214: “The differences in CT values between saliva (with or without RNA extraction) and NP swab were not statistically significant (Fig 1A), but CT values in saliva with vs. without RNA extraction are statistically different, with extraction of RNA consistently decreasing the CT values obtained.” To better understand the data, it would be useful to report the median values (IQR) of the Ct values obtained by the 3 detection methods.

->We have added the CT median values and the interquartile range of the 3 conditions to the text, as requested.

- How can the authors explain the fact that samples with RNA extraction had lower labours than those without extraction?

->We must apologize but we do not fully understand the question. We will answer assuming that the reviewer is querying about the CT value and not labour, as written in the question, but if not, please formulate the question in another way. A lower CT value corresponds to a higher RNA level being detected. When we extract RNA from samples, we are not only concentrating the RNA 4-fold (assuming no loss of RNA, which is incorrect), but also eliminating impurities that can interfere with the RT-qPCR reaction, so it is expected that the CT values are lower than those from the corresponding samples without extraction. We have added this explanation to the manuscript.

- Figure 1: Please report sample size

->We have added the sample size to the figure legend, as requested.

- Lines 228-230: “Next, we analyzed the stability of saliva samples stored for 3 days at 4ºC or 7 days at -20ºC, prior to processing (Fig 1B). We detected viral RNA in all saliva samples kept at 4ºC and -20ºC, regardless of RNA extraction, demonstrating they are stable under the conditions studied.” The detection of the virus has not changed, but if you look at the figure 1B, a clear increase in CT in samples not extracted and stored at -20 for 7 days can be appreciated. Have the authors defined if this increase is statistically significant?

Moreover, this may affect the outcome of the result. For example, if the sample processed immediately had 35 Ct there is a high risk that it will be negative after a -20 storage 7 days long.

->Yes, the increase in CT values after storage at -20ºC is statistically significant by Wilcoxon matched-pairs signed-rank test, with *p<0.05, compared to either the samples processed immediately or stored for 3 days at 4ºC, with or without RNA extraction. We have added the statistical analysis to figure 1B (and also to figure 2C)

->We agree with the reviewer that an adult sample with a CT value of 35 may be negative after -20ºC storage, and we have changed the text to include this observation. For that reason, we favour keeping saliva samples at 4ºC or at -80ºC. Importantly, we did not see an increase in CT values after saliva storage at -20ºC in the children population. 

- Table 2: Please report the % of Concurrent conditions.

->We have added the missing information to the table.

- Lines 257-260: “It is important to notice that there was a time interval between NP swab and saliva collection that could go up to 48 h and, therefore, these differences may not reflect a lower sensitivity of our method but rather a decrease in the patient viral load.” Stratifying the samples as: collected within 24 hours of TNF and collected 24-48 hours later, did you see differences in terms of CT values? And if so, did the former have values more similar to those obtained in TNF?

->We have done that analysis, but we could not find significant differences between the two populations.

- Figure 1 e 2: “Saliva stability at 4ºC and -20ºC: comparison of CT values from paired saliva samples (with and without RNA extraction) processed immediately, after 3 days at 4ºC or 7 days at -20ºC".” Non-extracted samples do not seem similar between adults and children. Is there an explanation for this?

->We thank the reviewer for this interesting observation that we had not paid attention to. One possible explanation for this discrepancy is that saliva composition is different in children and adults (2015, Abd; DOI: 10.21275/ART20164286). Different saliva composition may affect the stability of the RNA in the freezing/thawing process that the sample is subjected to in the -20ºC storage but not in the 4ºC storage. As we did not investigate this further, our explanation is just speculative and requires further research.

- Table 3: please define the populations used to calculate p-values.

->We have added the requested information to the table’s legend.

- No symptoms concerning the lower respiratory tract (pneumonia, bronchiolitis) were described in the sampled population. This does not allow to define the specificity and sensitivity of the salivary method in detecting SARS-CoV-2 in subjects developing lower respiratory tract infections respect to those developing upper respiratory tract symptoms only. This limitation should be mentioned in the discussion section.

->From what we understand from the reviewer’s comment, the premise is that sensitivity of the test in saliva would be different (lower?) in individuals with lower respiratory tract infections compared to upper respiratory tract infections. However, the adult population we tested, for which we had 100% sensitivity in saliva, consisted mainly in individuals with severe disease and lower respiratory tract infections. On the contrary, the majority of the tested children were asymptomatic or with mild symptoms associated with an upper respiratory tract infections. We think that this relates to the amount of viruses and their duration during the course of infection being much higher in case of severe disease.

---

## [Decision Letter · Decision Letter 1]

29 Apr 2022

Saliva molecular testing bypassing RNA extraction is suitable for monitoring and diagnosing SARS-CoV-2 infection in children

PONE-D-22-00630R1

Dear Dr. Amorim,

We’re pleased to inform you that your manuscript has been judged scientifically suitable for publication and will be formally accepted for publication once it meets all outstanding technical requirements.

Kind regards,

Ruslan Kalendar

Academic Editor

PLOS ONE

---

## [Editor Report · Acceptance letter]

19 May 2022

PONE-D-22-00630R1 

*Saliva molecular testing bypassing RNA extraction is suitable for monitoring and diagnosing SARS-CoV-2 infection in children*

Dear Dr. Amorim:

I'm pleased to inform you that your manuscript has been deemed suitable for publication in PLOS ONE. Congratulations! Your manuscript is now with our production department. 

Kind regards, 

on behalf of

Professor Ruslan Kalendar 

Academic Editor

PLOS ONE